# Amide Proton Transfer-Weighted Imaging Combined with ZOOMit Diffusion Kurtosis Imaging in Predicting Lymph Node Metastasis of Cervical Cancer

**DOI:** 10.3390/bioengineering10030331

**Published:** 2023-03-06

**Authors:** Qiuhan Huang, Yanchun Wang, Xiaoyan Meng, Jiali Li, Yaqi Shen, Xuemei Hu, Cui Feng, Zhen Li, Ihab Kamel

**Affiliations:** 1Department of Radiology, Tongji Hospital, Tongji Medical College, Huazhong University of Science and Technology, Wuhan 430030, China; 2Russell H. Morgan Department of Radiology and Radiological Science, The Johns Hopkins Medical Institutions, Baltimore, MD 21218, USA

**Keywords:** amide proton transfer-weighted imaging, diffusion kurtosis imaging, lymph node metastasis, cervical cancer

## Abstract

Background: The aim of this study is to investigate the feasibility of amide proton transfer-weighted (APTw) imaging combined with ZOOMit diffusion kurtosis imaging (DKI) in predicting lymph node metastasis (LNM) in cervical cancer (CC). Materials and Methods: Sixty-one participants with pathologically confirmed CC were included in this retrospective study. The APTw MRI and ZOOMit diffusion-weighted imaging (DWI) were acquired. The mean values of APTw and DKI parameters including mean kurtosis (MK) and mean diffusivity (MD) of the primary tumors were calculated. The parameters were compared between the LNM and non-LNM groups using the Student’s t-test or Mann–Whitney U test. Binary logistic regression analysis was performed to determine the association between the LNM status and the risk factors. The diagnostic performance of these quantitative parameters and their combinations for predicting the LNM was assessed with receiver operating characteristic (ROC) curve analysis. Results: Patients were divided into the LNM group (n = 17) and the non-LNM group (n = 44). The LNM group presented significantly higher APTw (3.7 ± 1.1% vs. 2.4 ± 1.0%, *p* < 0.001), MK (1.065 ± 0.185 vs. 0.909 ± 0.189, *p* = 0.005) and lower MD (0.989 ± 0.195 × 10^−3^ mm^2^/s vs. 1.193 ± 0.337 ×10^−3^ mm^2^/s, *p* = 0.035) than the non-LNM group. APTw was an independent predictor (OR = 3.115, *p* = 0.039) for evaluating the lymph node status through multivariate analysis. The area under the curve (AUC) of APTw (0.807) was higher than those of MK (AUC, 0.715) and MD (AUC, 0.675) for discriminating LNM from non-LNM, but the differences were not significant (all *p* > 0.05). Moreover, the combination of APTw, MK, and MD yielded the highest AUC (0.864), with the corresponding sensitivity of 76.5% and specificity of 88.6%. Conclusion: APTw and ZOOMit DKI parameters may serve as potential noninvasive biomarkers in predicting LNM of CC.

## 1. Introduction

Cervical cancer (CC) ranks as the most common gynecologic malignant cancer and one of the leading causes of cancer-specific death in women globally [1]. Lymph node metastasis (LNM) is a major prognostic indicator and an important determinant in treatment options in patients with CC [2]. For patients with early-stage CC (IA, IB1, IB2, and IIA1), radical hysterectomy with lymphadenectomy is commonly recommended [3]. Approximately 8–26% of patients with early stage cancer exhibit pathological LNM and require further postoperative chemo-radiotherapy [4]. If lymph node status is accurately diagnosed, patients can be managed medically, avoiding unnecessary invasive surgery [4].

MRI has been established as the main imaging modality in pre-treatment assessment of LNM in patients with CC [5]. The application of the conventional MRI sequences, which are mainly based on morphologic features, is still challenging due to their low sensitivity (29–69%) for assessing the presence of LNM [6,7]. Diffusion-weighted imaging (DWI) has been successfully introduced as a non-invasive technique to reveal tissue microstructural changes in vivo [8]. ZOOMit DWI applies echo-planar imaging and another parallel radiofrequency pulse sequence to obtain a zoomed field-of-view (FOV) that only covers the region of interest (ROI) and consequently reduces geometric distortion and susceptibility artifacts, which allows for better image quality and more anatomical detail [9]. ADC is a quantitative biomarker that has been widely utilized in oncologic applications. However, no single ADC value has been established as an indicator of lymph node positivity in CC [10]. The non-Gaussian diffusion model—diffusion kurtosis imaging (DKI)—has the potential in quantifying the microstructural heterogeneity of tissues [11,12]. Several previous studies indicated that the DKI-derived parameters were able to estimate the histological features and predict the curative response of CC [13,14,15]. It has been reported that tumors with higher heterogeneity are more prone to lymph node metastasis, suggesting that DKI may be useful in predicting the LNM based on the primary tumors.

Furthermore, chemical exchange saturation transfer (CEST) imaging provides more metabolic information than the changes of tissue microstructure [16]. Amide proton transfer-weighted (APTw) MRI, a subtype of chemical exchange saturation transfer imaging, is a molecular MRI technique that mainly explores the chemical transfer properties of amide protons located at a chemical shift of 3.5 ppm [17]. The APT signal intensity (APT SI) reflects the concentrations of mobile macromolecules, such as proteins and peptides. With the ability to obtain biochemical information, APTw enables the exploration of tissue microenvironment noninvasively, and has gained increasing interest as a valuable adjunct to conventional MRI [18]. Previous studies demonstrated that APTw MRI has been successfully applied to brain tumor [19], breast cancer [20], prostate cancer [21], rectal cancer [22], bladder cancer [23], etc.

APTw and DKI offer the ability for visualizing the microenvironment and microstructural heterogeneity of tumor tissues, reportedly linked to tumor characteristics, thus with the potential for predicting LNM. So far, few studies focused on estimating the presence of LNM. In this study, we aimed to investigate the potential of the combination of APTw MRI with ZOOMit DKI in predicting the pretreatment LNM in CC based on the primary tumors.

## 2. Materials and Methods

### 2.1. Patient Population

The ethics committee of our hospital approved this retrospective study, and the requirement for informed consent was waived. From September 2021 to July 2022, a consecutive series of 105 patients suspected of having CC were enrolled in this cohort and underwent pelvic MRI. The inclusion criteria were as follows: (1) no therapy performed prior to the MRI examination and (2) no contraindications to MRI. The exclusion criteria were as follows: (1) no or incomplete histopathological results (n = 21); (2) confirmed non-cervical cancer (n = 9); (3) maximum tumor diameter less than 10 mm (n = 10); and (4) inadequate image quality due to major artifacts (n = 4). Ultimately, a total of 61 patients with pathologically diagnosed CC were included in this study. The mean age of the patients was 51 ± 12 years (range, 28–78 years). Surgery was performed within 14 days of the MRI examination. The flowchart of the patient selection process is shown in Figure 1.

### 2.2. MR Imaging Protocol

All participants underwent MR on a 3.0-T MRI scanner (Magnetom Skyra, Siemens Healthcare, Erlangen, Germany) with an 18-channel body phased-array coil. All patients were instructed not to urinate for at least 1 h before the MRI examination so that the bladder would be moderately distended during image acquisition. The routine scan sequences including the T1- and T2-weighted imaging were performed.

CEST-MRI was provided by Zhang Yi’s team of Zhejiang University [24]. The middle slice of axial CEST images was located through the maximum cross section of the tumor present on T2WI images. In the CEST saturation module, ten Gaussian saturation pulses were applied with each a duration of 100 ms and a saturation power level of 2.0 micro T. A total of 63 frequency offsets were acquired for the Z-spectrum. The 63 offsets included reference image and saturated scans at 80, 70, 60, 50, 40, 30, 20, 15.625, 10, ±6, ±5, ±4.5, ±4 (2), ±3.75 (2), ±3.5 (6), ±3.25 (2), ±3 (2), ±2.5 (2), ±2 (2), ±1.5, ±1, ±0.75, ±0.5, ±0.25, and 0 ppm, where numbers in parentheses represented the number of repetitions [24,25]. In addition, the gradient echo (GRE) sequence was applied for B0 inhomogeneity correction. The B0 map was calculated as the division of the phase difference between GRE phase images acquired with two different TEs to the TE difference of 4.92 ms. Then, for each pixel, the Z-spectrum is shifted using the B0 previously calculated. Using B0-corrected magnetization transfer ratio asymmetry (MTRasym) at 3.5 ppm offset, the APTw image was computed. The other parameters were as follows: repetition time (TR)/echo time (TE) = 3100/7.1 ms, field of view = 380 × 332 mm^2^, matrix size = 128 × 128, slice thickness = 4 mm, inter-slice gap = 1 mm, bandwidth = 399 Hz, and acquisition time = 3 min 20 s.

Axial ZOOMit diffusion-weighted images (DWI) with *b*-values of 0, 500, 1000, 1500, and 2000 s/mm^2^ were obtained using a single-shot spin-echo echo-planar imaging sequence. The other acquisition parameters for the DWI sequence were TR/TE = 8000/69.5 ms, field of view = 240 × 100 mm^2^, matrix size = 120 × 120, slice thickness = 4 mm, inter-slice gap = 1 mm, bandwidth = 1666 Hz, and acquisition time = 4 min 30 s.

### 2.3. Image Analysis

All data were transferred to post-processing software for quantitative analysis. APTw images were analyzed using MATLAB software (The MathWorks, Inc., Natick, MA, USA) based on the original Z-spectral signal [24,25]. The APTw parameter was computed as the MTRasym at 3.5 ppm, which was calculated using the following equation:(1)$${\mathrm{MTR}}_{\mathrm{asym}}\;\left(3.5\;\mathrm{ppm}\right)={\;S}_{\mathrm{sat}}\;\left(-3.5\;\mathrm{ppm}\right)/S_{0}-{\;S}_{\mathrm{sat}}\;\left(+3.5\;\mathrm{ppm}\right)/S_{0}$$
where MTRasym (3.5 ppm) is the asymmetric magnetization transfer rate at 3.5 ppm, S_sat_ represents signal intensity (SI) obtained with applied saturation pulse, and S_0_ represents SI with unsaturated pulse.

The ZOOMit DWI images were processed using MR Body Diffusion Toolbox v1.5.0 (Siemens Healthcare, Erlangen, Germany). For the DKI model, DKI parameters were calculated using the following equation [26] with five *b*-values (0, 500, 1000, 1500, and 2000 s/mm^2^):(2)$$S_{b}={\;S}_{0}\times \exp\left(-b\times \;\mathrm{MD}\;+{\;b}^{2}\times {\;\mathrm{MD}}^{2}\times \mathrm{MK}/6\right)$$
where S_b_ is the SI at a particular *b* value, S_0_ is the SI when *b* = 0 s/mm^2^, mean kurtosis (MK) indicates the degree of dispersion deviation from Gaussian distribution, and mean diffusivity (MD) represents the apparent diffusion coefficient (ADC) after non-Gaussian behavior modification.

The MRI images were analyzed by two radiologists (Y.W. and C.F., with 10 and 15 years of experience in pelvic MRI diagnosis, respectively) independently, blinded to the histopathologic findings. The regions of interest (ROIs) were manually delineated along the tumor border on the largest cross-sectional tumor area on APTw images and on DKI maps, using the T2WI as a reference (Figure 2 and Figure 3), while carefully avoiding the areas of necrosis, cystic degeneration, hemorrhage, and blood vessels. The average value of each parameter as measured by the two radiologists was taken for final analysis.

### 2.4. Statistical Analysis

Statistical analysis was performed using SPSS (version 23.0; IBM Corp., Armonk, N.Y., USA), MedCalc (version 20.0; MedCalc Software Ltd., Ostend, Belgium), and GraphPad Prism (version 9.0; GraphPad Software, San Diego, CA, USA). Intraclass correlation coefficients (ICCs) were used to assess the interobserver agreement of each parameter (<0.4, low consistency; 0.40–0.75, medium consistency; >0.75, high consistency). Kolmogorov–Smirnov test was utilized to estimate the normality of the continuous variables. Continuous variables were expressed as mean ± standard deviation (SD) and were compared between LNM and non-LNM groups with Student’s t-test or Mann–Whitney U test. Categorical variables were expressed as counts and percentages and compared using χ2 test or Fisher’s probability analysis. Binary logistic regression analysis was performed to evaluate the association between the LNM status and the risk factors, and to determine diagnostic efficacy of combined parameters. Receiver operating characteristic curve (ROC) analysis was used to evaluate the diagnostic performance of CEST and DKI parameters for discriminating LNM from non-LNM. Meanwhile, the area under curve (AUC), sensitivity, and specificity were reported by using the Youden index. AUCs of different parameters were compared using the DeLong test. Spearman correlation coefficients were calculated between the metrics and lymph node status. The two-sided *p* < 0.05 was considered statistically significant.

## 3. Results

### 3.1. Participant Characteristics

The clinicopathologic data of the 61 included patients are presented in Table 1. The participants were divided into LNM group (n = 17) and non-LNM group (n = 44), according to the pathological findings. Statistically significant differences in tumor size, FIGO stage, histological grade, and depth of invasion were found between the LNM and non-LNM groups (*p* = 0.009, *p* < 0.001, *p* = 0.009, and *p* < 0.001, respectively). There were no significant differences in age, menopausal status, histological classification, squamous cell carcinoma antigen (SCC-Ag) level, and vascular invasion between the LNM and non-LNM groups (all *p* > 0.05).

### 3.2. Interobserver Reliability Analysis

The interobserver consistency assessment of APTw and DKI-derived parameters shows excellent reproducibility, with ICCs of 0.908 (95% CI: 0.855, 0.943), 0.987 (95% CI: 0.979, 0.992), and 0.984 (95% CI: 0.974, 0.990) for APTw, MK, and MD, respectively.

### 3.3. Comparisons of APTw, MK, and MD between Different Histopathological Parameters

Comparisons of APTw, MK, and MD between different histopathological parameters are shown in Table 2. The LNM group presented significantly higher APTw (3.7 ± 1.1% vs. 2.4 ± 1.0%, *p* < 0.001) and MK (1.065 ± 0.185 vs. 0.909 ± 0.189, *p* = 0.005) and lower MD (0.989 ± 0.195 × 10^−3^ mm^2^/s vs. 1.193 ± 0.337 × 10^−3^ mm^2^/s, *p* = 0.035) than the non-LNM group. The comparisons of APTw, MK, and MD between LNM and non-LNM groups are shown in Figure 4. The APTw of the tumor was also significantly higher in advanced-stage cervical cancer (FIGO stage IIB-IV) and deeper invasion (≥2/3 of cervical wall) (*p* = 0.002 and *p* = 0.001, respectively). The value of MK was higher in patients with a larger tumor size (≥4 cm), advanced-stage cervical cancer, higher histological grade, and deeper invasion (*p* = 0.009, *p* < 0.001, *p* = 0.016, and *p* < 0.001, respectively). Furthermore, tumors with a larger tumor size, CSC group, advanced-stage cervical cancer, and deeper invasion presented lower MD values (*p* = 0.005, *p* = 0.014, *p* < 0.001, and *p* = 0.011, respectively).

### 3.4. Univariate and Multivariate Logistic Regression Analyses

The univariate and multivariate logistic regression analyses for assessing the indicators of LNM are shown in Table 3. In univariate analysis, the APTw (OR = 3.523, *p* = 0.001), MK (OR = 1.005, *p* = 0.011), MD (OR = 0.997, *p* = 0.029), tumor size (OR = 1.041, *p* = 0.016), histological grade (OR = 5.687, *p* = 0.008), and depth of invasion (OR = 23.111, *p* = 0.003) were significantly correlated with the lymph node status of cervical cancer. There were no significant correlations between the lymph node status and age, menopausal status, histological classification, SCC-Ag level, and vascular invasion. Multivariate logistic analysis revealed that APTw (OR = 3.115, *p* = 0.039) and depth of invasion (OR = 25.473, *p* = 0.031) were independent predictors for evaluating the status of LNM in CC. To make the model more readable, a nomogram based on predictive logistic regression model was developed to make the results more intuitive. The nomogram is shown as Figure 5.

### 3.5. Diagnostic Performance of the APTw, MK, and MD in Predicting Lymph Node Status

The diagnostic performance of APTw and DKI-derived parameters (MK and MD) and their combinations for discriminating LNM from non-LNM are shown in Table 4 and Figure 6. The AUC (0.807) of APTw was higher than that of MK (AUC, 0.715) and that of MD (AUC, 0.675) for discriminating LNM from non-LNM, but the differences were not significant (all *p* > 0.05). Moreover, the combination of APTw, MK, and MD yielded the highest AUC (0.864), with the corresponding sensitivity of 76.5% and specificity of 88.6%. Significantly higher AUC (0.864) was observed in the combination of APTw, MK, and MD than in MD alone (0.675; *p* = 0.010) for differentiating the lymph node status.

## 4. Discussion

Our results demonstrated that APTw outperformed the ZOOMit DKI parameters MK and MD in predicting the LNM of CC, and the diagnostic performance could be further improved by the combination of APTw, MK, and MD. With these results, our study indicated that the combination of APTw and ZOOMit DKI could be used as a potential non-invasive biomarker to predict the LNM of CC

The DKI model reflects the non-Gaussian diffusion property caused by the microstructural complexity of tissues and therefore has the potential in quantifying the microstructural heterogeneity of tissues [11,12]. The DKI parameter MK is reported to positively correlate with the heterogeneity of tissue microstructure, while MD represents how freely water can diffuse through a tissue, with lower MD indicating impaired diffusion and probably denser tissue [27]. In this study, CC with higher tumor grade and advanced FIGO stage presented significantly higher MK and lower MD. It is presumed that tumors with higher heterogeneity are more prone to lymph node metastasis, suggesting that DKI parameters may be useful in predicting the LNM based on the primary tumors. Due to the difficulty of matching the lymph nodes on images to the pathologic findings [28], our study attempted to investigate the lymph node status based on the primary tumors, rather than analyzing the lymph nodes directly. Our finding demonstrated that LNM group showed significantly higher MK value and lower MD value than non-LNM group. A similar result was also reported by Yamada et al. [14,29], suggesting that the tumors with LNM had more complex tissue microstructures, which can further limit the diffusion of water molecules [30].

With the ability to detect the exchange of the amide protons in protein or polypeptide with hydrogen protons in water [17], APTw imaging has been successfully applied to characterize rectal cancers [15], bladder tumors [23], endometrial carcinomas [17], and cervical cancers [30]. In this study, higher APTw values were found in CC with higher FIGO stage and deeper tumor invasion, further indicating the possible feasibility of APTw in characterizing cervical cancers. Furthermore, this study indicated that the tumors with LNM presented higher APTw values than the tumors with non-LNM. A similar result was also previously reported in rectal adenocarcinoma by Chen et al. [15]. The possible reason may be that higher levels of proliferation require enhanced protein synthesis, resulting in accumulation of intra-cellular proteins. Hence, higher mobile protein and peptide concentrations of tumor in LNM group might be the main reason for higher APTw in CC. Furthermore, Meng et al. [30,31] demonstrated that APTw value of high grade CC was significantly higher than that of low grade CC, and a similar finding was also reported in bladder cancer [23]. However, a significant difference was not observed between the low grade and high grade CC in this study. The reason may be that only the patients with CSC were included in previous studies [27,32,33], while in this study, the patients with CA were not excluded. CA originates from endocervical cells and therefore has rich glandular structure and the ability to secrete mucin. Due to the different cell origins, APTw values between CSC and CA could be quite different, which may result in inconsistencies with the results of previous studies.

In this study, the APTw, MK, MD, tumor size, histological grade, and depth of invasion were significantly correlated with the lymph node status of CC in univariate analysis. The CC with LNM exhibits more aggressive biological behavior, which is most likely present with larger tumor size, high cellular density, rapid cell proliferation, and enhanced metabolism [34]. The APTw was an independent predictor for LNM of CC in multivariable analysis, suggesting the potential role of APTw in predicting the LNM of CC.

Our results demonstrated that the AUC of APTw (0.807) was higher than those of MK (0.715) and MD (0.675) in differentiating LNM from non-LNM, suggesting the advantage of APTw over the DKI parameters in predicting the LNM of CC. The possible reason may be that the DKI parameters were derived based on the high *b*-value (2000 s/mm^2^) DWI, which is more susceptible to the effects of low signal-to-noise ratio and image distortion, possibly resulting in measurement bias [35]. The diagnostic performance of the combination of APTw and DKI parameters for predicting the LNM of CC was also assessed in this study, attempting to explore the changes of metabolic information and the heterogeneity of tumor microstructure simultaneously. The combination of APTw, MK, and MD yielded the highest AUC (0.864) in discriminating LNM from non-LNM, indicating the added value of APTw to DKI in predicting LNM of CC based on the primary tumors.

There are some limitations in this study. First, the sample size of our study was relatively small, resulting in insufficient detection of statistical significance for some variables, such as tumor grade. Second, this study was a single-center study, and it has potential selection bias. Third, only single-slice two-dimensional images of APTw MRI were obtained, rather than three-dimensional images, thus it was impossible to extract the information of the whole tumors.

## 5. Conclusions

In conclusion, both APTw and ZOOMit DKI parameters have the potential to predict LNM of CC, and the diagnostic performance is further improved by combining both parameters. The APTw and ZOOMit DKI could be used as promising non-invasive tools to predict the LNM of CC, thus aiding in tailoring treatment modality for patients with CC.

## Figures and Tables

**Figure 1 bioengineering-10-00331-f001:**
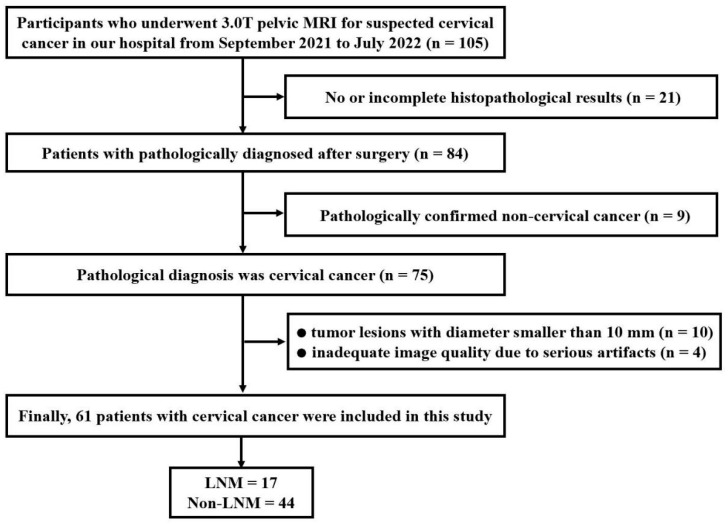
Flow chart of patient enrollment.

**Figure 2 bioengineering-10-00331-f002:**
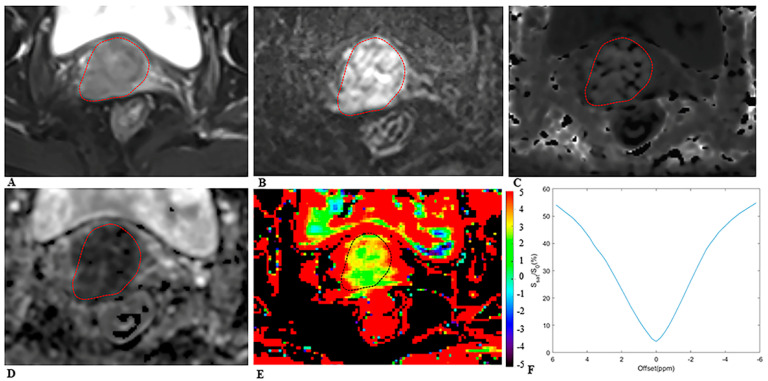
MRI scans in a 64−year−old woman with cervical cancer and postoperative pathology revealed LNM. Axial (**A**) T2−weighted image illustrates an exophytic tumor on the cervix wall. A diffusion−weighted image (**B**) with *b* = 1000 s/mm^2^ shows a high−signal−intensity tumor. Mean kurtosis (MK) map (**C**) and mean diffusivity (MD) map (**D**) generated from DKI model. Amide proton transfer−weighted (APTw) image (**E**) and Z−spectrum of the tumor (**F**); the color bar indicates the APTw value. The mean MK, MD, and APTw values measured by the two radiologists were 1.116, 0.774 × 10^−3^ mm^2^/s, and 4.5%, respectively.

**Figure 3 bioengineering-10-00331-f003:**
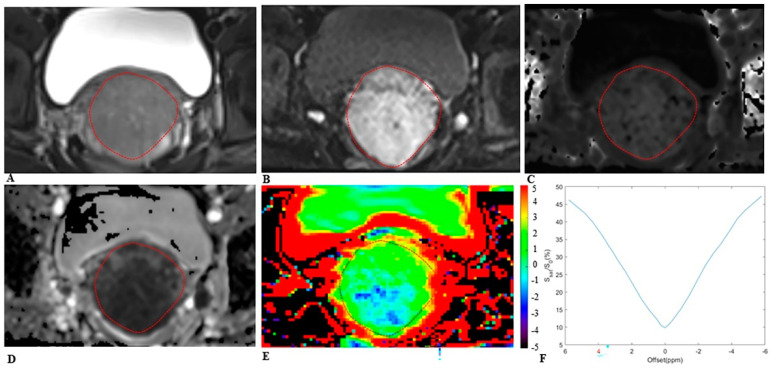
MRI scans in a 25−year−old woman with cervical cancer and postoperative pathology revealed non−LNM. Axial (**A**) T2−weighted image illustrates an exophytic tumor on the cervix wall. A diffusion−weighted image (**B**) with *b* = 1000 s/mm^2^ shows a high−signal−intensity tumor. Mean kurtosis (MK) map (**C**) and mean diffusivity (MD) map (**D**) generated from DKI model. Amide proton transfer−weighted (APTw) image (**E**) and Z−spectrum of the tumor (**F**); the color bar indicates the APTw value. The mean MK, MD, and APTw values measured by the two radiologists were 0.832, 1.123 × 10^−3^ mm^2^/s, and 1.7%, respectively.

**Figure 4 bioengineering-10-00331-f004:**
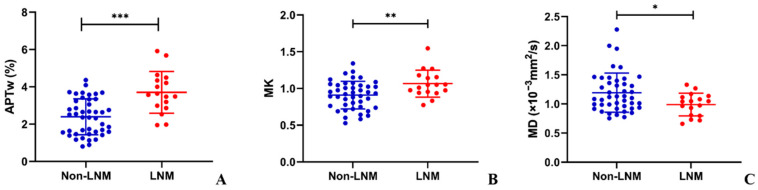
The scatter plots of (**A**) amide proton transfer-weighted (APTw), (**B**) mean kurtosis (MK), and (**C**) mean diffusivity (MD) values between LNM and non-LNM groups in cervical cancer. There were significant differences in APTw, MK, and MD values between LNM group and non-LNM group (* *p* < 0.05; ** *p* < 0.01; *** *p* < 0.001).

**Figure 5 bioengineering-10-00331-f005:**
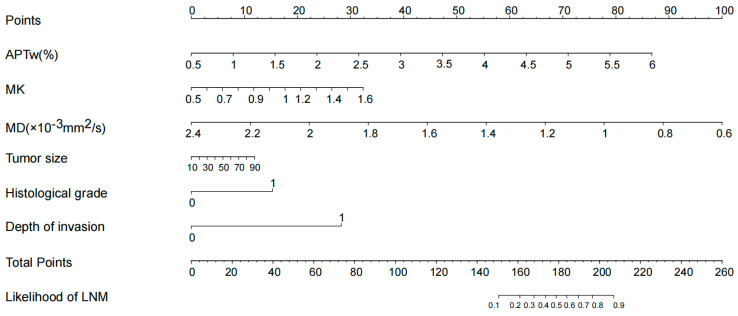
According to the logistic regression model, the nomogram was obtained. The probability of LNM was calculated by substituting various factors into the nomogram.

**Figure 6 bioengineering-10-00331-f006:**
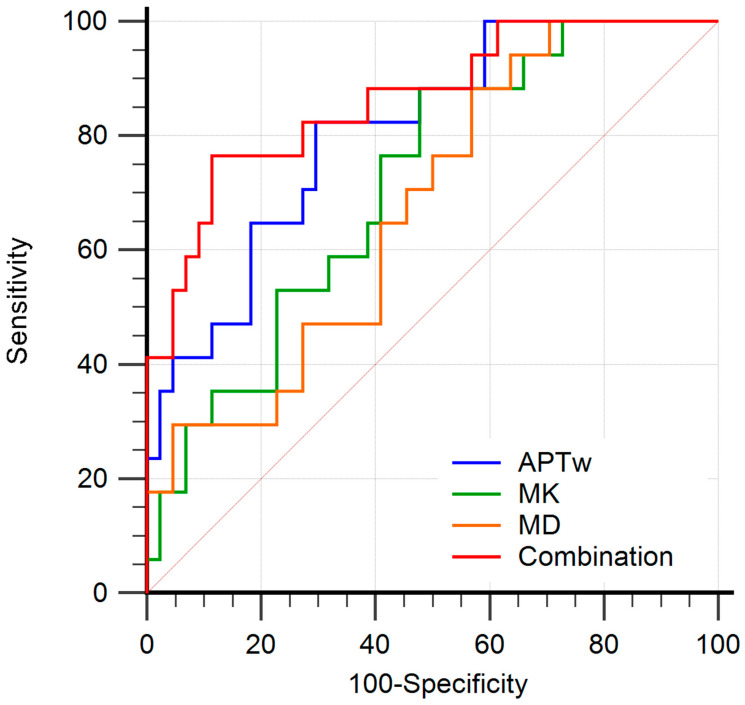
Receiver operating characteristic curves (ROC) analysis of each parameter for predicting the LNM in patients with cervical cancer. The area under curves (AUCs) of amide proton transfer-weighted (APTw), mean kurtosis (MK), mean diffusivity (MD) values, and combination model (APTw + MK + MD) to evaluate LNM status were 0.807 (95% CI: 0.686–0.897), 0.715 (95% CI: 0.585–0.823), 0.675 (95% CI: 0.543–0.790), and 0.864 (95% CI: 0.752–0.938), respectively. The combination of APTw, MK, and MD yielded the highest AUC.

**Table 1 bioengineering-10-00331-t001:** Clinicopathological characteristics and lymph node status of patients with cervical cancer.

Characteristics		Total	Non-LNM	LNM	*p*
No. of Patients		61	44	17	
Age (years) ^a^		51 ± 12 (28–78)	50 ± 12 (28–78)	52 ± 12 (29–70)	0.513
Tumor size (mm) ^a^		36 ± 21 (13–105)	31 ± 17 (15–72)	44 ± 19 (13–105)	**0.009**
Menopausal status					0.437 *
	No	30 (49.2)	23 (52.3)	7 (41.2)	
	Yes	31 (50.8)	21 (47.7)	10 (58.8)	
Histological classification					0.423 *
	CSC	52 (85.2)	36 (81.8)	16 (94.1)	
	CA	9 (14.8)	8 (18.2)	1 (5.9)	
FIGO stage					**<0.001 ***
	Early stage (IB-IIA)	32 (52.5)	31 (70.5)	1 (5.9)	
	Advanced stage (IIB-IV)	29 (47.5)	13 (29.5)	16 (94.1)	
Histologic grade					**0.009 ***
	Low grade	32 (52.5)	28 (63.6)	4 (23.5)	
	High grade	29 (47.5)	16 (36.4)	13 (76.5)	
Depth of invasion					**<0.001 ***
	<2/3 of cervical wall	27 (44.3)	26 (59.1)	1 (5.9)	
	≥2/3 of cervical wall	34 (55.7)	18 (40.9)	16 (94.1)	
SCC-Ag level					0.133 *
	≤1.5 ng/mL	21 (34.4)	18 (40.9)	3 (17.6)	
	>1.5 ng/mL	40 (65.6)	26 (59.1)	14 (82.4)	
Vascular invasion					0.421 *
	No	44 (72.1)	33 (75.0)	11 (64.7)	
	Yes	17 (27.9)	11 (25.0)	6 (35.3)	

^a^ Numbers are means ± standard deviations with ranges in parentheses; other data are presented as n (%). * Chi-square test used. The bold font in the table indicates the comparison with statistical significance. LNM = lymph node metastasis; Non-LNM = non-lymph node metastasis; CSC = cervical squamous carcinoma; CA = cervical adenocarcinoma; FIGO = International Federation of Gynecology and Obstetrics; SCC-Ag = squamous cell carcinoma antigen.

**Table 2 bioengineering-10-00331-t002:** Comparison of APTw and DKI-derived parameters among different histopathological parameters of cervical cancer.

Variables		APTw (%)	*p*	MK	*p*	MD (×10^−3^ mm^2^/s)	*p*
Lymph node status			**<0.001**		**0.005**		**0.035 ***
	Non-LNM (n = 44)	2.4 ± 1.0		0.909 ± 0.189		1.193 ± 0.337	
	LNM (n = 17)	3.7 ± 1.1		1.065 ± 0.185		0.989 ± 0.195	
Tumor size			0.319		**0.009**		**0.005 ***
	<4 cm (n = 39)	2.6 ± 1.2		0.903 ± 0.191		1.219 ± 0.336	
	≥4 cm (n = 22)	3.0 ± 1.1		1.040 ± 0.187		0.990 ± 0.213	
Histological classification			0.447		0.471		**0.014 ***
	CSC (n = 52)	2.8 ± 1.2		0.960 ± 0.191		1.076 ± 0.230	
	CA (n = 9)	2.5 ± 1.2		0.908 ± 0.247		1.485 ± 0.501	
FIGO stage			**0.002**		**<0.001**		**<0.001 ***
	Early-stage (n = 32)	2.3 ± 1.0		0.857 ± 0.186		1.285 ± 0.345	
	Advanced-stage (n = 29)	3.2 ± 1.1		1.058 ± 0.157		0.973 ± 0.171	
Histological grade			0.117		**0.016**		0.142 *
	Low grade (n = 32)	2.5 ± 1.0		0.895 ± 0.195		1.193 ± 0.371	
	High grade (n = 29)	3.0 ± 1.2		1.016 ± 0.187		1.074 ± 0.232	
Depth of invasion			**0.001**		**<0.001**		**0.011 ***
	<2/3 of cervical wall (n = 27)	2.2 ± 0.9		0.845 ± 0.175		1.270 ± 0.381	
	≥2/3 of cervical wall (n = 34)	3.2 ± 1.2		1.038 ± 0.176		1.031 ± 0.202	
Vascular invasion			0.140		0.210		0.394 *
	No (n = 44)	2.6 ± 1.0		0.933 ± 0.215		1.132 ± 0.345	
	Yes (n = 17)	3.1 ± 1.4		1.004 ± 0.142		1.148 ± 0.234	

* Mann–Whitney U test; others are Students’ t-test. Bold type face in the table indicates that the comparison is statistically significant. APTw = amide proton transfer-weighted; MK = mean kurtosis; MD = mean diffusivity; LNM= lymph node metastasis; Non-LNM = non-lymph node metastasis; CSC = cervical squamous carcinoma; CA = cervical adenocarcinoma; FIGO = International Federation of Gynecology and Obstetrics.

**Table 3 bioengineering-10-00331-t003:** Univariate and multivariate logistic regression analysis for LNM in cervical cancer.

Variable	Univariate Analysis		Multivariate Analysis	
	Odds Ratio (95 % CI)	*p*	Odds Ratio (95 % CI)	*p*
APTw (%)	3.523 (1.676, 7.404)	**0.001**	3.115 (1.059, 9.162)	**0.039**
MK	1.005 (1.001, 1.008)	**0.011**	1.000 (0.991, 1.008)	0.911
MD (×10^−3^ mm^2^/s)	0.997 (0.994, 1.000)	**0.029**	0.998 (0.990, 1.005)	0.503
Age	1.017 (0.968, 1.068)	0.507		
Tumor size	1.041 (1.008, 1.075)	**0.016**	0.949 (0.878, 1.025)	0.184
Menopausal status	1.565 (0.504, 4.856)	0.439		
Histological classification	0.281 (0.032, 2.440)	0.250		
Histologic grade	5.687 (1.585, 20.414)	**0.008**	1.628 (0.207, 12.781)	0.643
Depth of invasion	23.111 (2.808, 190.202)	**0.003**	25.473 (1.351, 480.376)	**0.031**
SCC-Ag level	3.231 (0.809, 12.896)	0.097		
Vascular invasion	1.636 (0.490, 5.467)	0.424		

All factors with *p* < 0.05 in univariate analysis were included in multivariate regression analysis. The bold typeface in the table indicates the logistic regression analysis with statistical significance. CI = confidence interval; LNM = lymph node metastasis; APTw = amide proton transfer-weighted; MK = mean kurtosis; MD = mean diffusivity; SCC-Ag = squamous cell carcinoma antigen.

**Table 4 bioengineering-10-00331-t004:** Diagnostic performance of APTw, MK, and MD values in predicting LNM in cervical cancer.

Parameters	Cutoff	AUC (95% CI)	Sensitivity (%)	Specificity (%)	*p*	*p* for Comparison
APTw (%)	2.856	0.807 (0.686–0.897)	82.4	70.5	<0.001	0.132
MK	0.932	0.715 (0.585–0.823)	88.2	52.3	0.002	0.053
MD (×10^−3^ mm^2^/s)	1.171	0.675 (0.543–0.790)	88.2	43.2	0.017	**0.010**
Combination	-	0.864 (0.752–0.938)	76.5	88.6	<0.001	Ref

Combination represents APTw + MK + MD. The bold typeface in the table indicates significant difference compared with the Ref by DeLong test. APTw = amide proton transfer-weighted; MK = mean kurtosis; MD = mean diffusivity; LNM = lymph node metastasis; CC = cervical cancer; AUC = area under the curve; Ref = reference.

## Data Availability

The datasets generated or analyzed during the study are available from the corresponding author upon reasonable request.

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
