# Peer review of "Amide Proton Transfer-Weighted Imaging Combined with ZOOMit Diffusion Kurtosis Imaging in Predicting Lymph Node Metastasis of Cervical Cancer"

_bioengineering, 2023, doi:10.3390/bioengineering10030331_

Round 1

Reviewer 1 Report

This an interesting systematic study aimed at combining amide proton transfer-weighted imaging with ZOOMit diffusion imaging for prediction of lymph node metastasis in cervical canser. It is well-written and clearly presented. I have only minor corrections to consider:

1. To my taste, ZOOMit was not really presented well in the introductory part. A couple of sentences more about advantages of this method could help the reader to understand imprortance of the results.

2. It seems that on-line the version of abstract is slightly different from the one in the PDF. This must be checked to have the latest version in both places. 

Author Response

Response to Reviewer 1 Comments

Point 1:To my taste, ZOOMit was not really presented well in the introductory part. A couple of sentences more about advantages of this method could help the reader to understand imprortance of the results.

Response 1: We thank the Reviewer’s suggestion. We have added a couple of sentences more about advantages of ZOOMit in the introductory part. The corresponding sentences are “ZOOMit DWI applied echo-planar imaging and another parallel radiofrequency pulse sequence to obtain a zoomed field-of-view (FOV) that only covers the region of interest (ROI) and consequently reduces geometric distortion and susceptibility artifacts, which allows for better image quality and more anatomical detail [1].

  1. Li, S.; He, K.; Yuan, G.; Yong, X.; Meng, X.; Feng, C.; Zhang, Y.; Kamel, I.R.; Li, Z. WHO/ISUP grade and pathological T stage of clear cell renal cell carcinoma: value of ZOOMit diffusion kurtosis imaging and chemical exchange saturation transfer imaging. Eur Radiol 2022, doi:10.1007/s00330-022-09312-2.

Point 2: It seems that on-line the version of abstract is slightly different from the one in the PDF. This must be checked to have the latest version in both places. 

Response 2: We thank the Reviewer’s suggestion. We have checked the latest version. The final version is subject to the revised version.

Reviewer 2 Report

1. This manuscript focuses on the combination of APT and kurtosis weighted MRI for the prediction of lymph node met statuses in a cohort of patients. It’s an interesting study looking to see how much information can be gained from the use of two different MRI contrasts that are sensitive to different biophysical processes, and combines them well.    

2. This is a relatively novel combination of MRI contrasts to help predict lymph node mets in cervical cancer. Both are available on commercial systems, and could be used to help in radiological decision making.  

Weaknesses: Needs further validation in a larger cohort / other centres.  

3. The manuscript is sound and reads well. No further improvements required from me than the ones I have already stated.

This is a well performed single center cohort study. I have no comments to improve the study design or conclusions. 

Author Response

Response to Reviewer 2 Comments

  1. This manuscript focuses on the combination of APT and kurtosis weighted MRI for the prediction of lymph node met statuses in a cohort of patients. It’s an interesting study looking to see how much information can be gained from the use of two different MRI contrasts that are sensitive to different biophysical processes, and combines them well.    
  2. This is a relatively novel combination of MRI contrasts to help predict lymph node mets in cervical cancer. Both are available on commercial systems, and could be used to help in radiological decision making.  

Weaknesses: Needs further validation in a larger cohort / other centres.  

  1. The manuscript is sound and reads well. No further improvements required from me than the ones I have already stated.

This is a well performed single center cohort study. I have no comments to improve the study design or conclusions. 

Response: Thanks for the reviewers' recognition of our article. We will validate our study in a larger cohort / other centers in future studies.

Reviewer 3 Report

Bioengineering-2179748: Amide Proton Transfer-weighted Imaging Combined with ZOOMit Diffusion Kurtosis Imaging in Predicting Lymph Node Metastasis of Cervical Cancer

Summary: The authors present a study of 61 patients with cervical cancer to examine the applicability of APT-weighted imaging and diffusion kurtosis imaging in predicting lymph node metastasis. They found that both methods correlated with lymph node status. However, they did not provide a hypothesis for their study, nor did they explain whether their finding would justify the increased imaging costs.

Major weaknesses of the paper include the fact that there is no hypothesis presented. The discussion focuses solely on the correlation between the imaging metrics and the lymph node status without addressing how useful either technique would be clinically. It is not clear how the lymph node status was determined. Assuming that the status was determined via pathology, it is not clear if the addition of these imaging methods would eliminate the need for pathological examination.

Specific comments:

1.      Page 2, line 54. ZOOMit is a Siemens-specific sequence. It would be helpful to explain what this sequence is in generic terms instead of using a vendor specific name, at least in the first mention.

2.      Page 2, general observation. There is no hypothesis presented nor is there any mechanistic explanation for why APT-weighted imaging or diffusion kurtosis would be any better than straight diffusion-weighted imaging.

3.      Page 2, line 82. This is obviously a research sequence, so the patients should have provided written informed consent for this. Even if the current analysis was on previously obtained data, such that informed consent was not needed for this specific analysis, some mention of informed consent for the original image acquisition should be made.

4.      Page 3, line 101. This reference cites a Philips scanner, but the current paper uses a Siemens scanner. Thus, someone did a conversion from one vendor to the other, which should be indicated in the description. This wasn't a binary code that was provided for plug 'n' play.

5.      Page 3: lines 105-113. What sort of readout sequence was used with the CEST imaging? Was it a gradient recalled echo, a fast spin echo, or an echo-planar readout? If it was a fast spin echo, how long was the echo train? If it was an EPI readout, was it single shot? Was fat sat used?  Is the bandwidth cited the bandwidth for excitation or readout? Or was it the bandwidth of the CEST pulse? Does the acquisition time refer to the entire set of 63 offsets? If so, then this almost has to be an EPI readout. Given that so many frequency offsets were acquired, why did the authors only use the simple MTRasym analysis instead of any kind of fitting (e.g. Zaiss’s method, give reference).

6.      Page 3, line 118: Is this bandwidth for the excitation pulse or the readout? And is the acquisition time the total required for all 5 b-values? How many directions were used?

7.      Page 4, figure 2. It would be helpful to outline the lesion on the T2 image and carry that outline across the other images. Why does the Z-spectrum run from 80 to -10 ppm? It would be much easier to assess the APT effect if the spectrum were only shown from about 8 to -8 ppm.

8.      Page 5, figure 3. Same comments as for previous figure.

9.      Page 5, line 186. Delete “while”, as it makes this sentence incomplete.

10.   Page 7, line 224. Delete "while”.

11.   Page 7, lines 220-223. If all of these parameters (tumor size, histological grade, depth of invasion) correlate with lymph node status, what is the advantage of adding APT and DKI? This needs to be addressed somewhere and there should be a huge advantage to offset the increased cost.

12.   Figure 4 shows quite a bit of overlap between the groups for the various MRI parameters. What does this mean for an individual patient? This should be discussed.

13.   Page 10, line 270. What do the authors mean by “blocking degree”? MD should represent how freely water can diffuse through a tissue, with lower MD indicating impaired diffusion and probably denser tissue.

14.   Page 10, line 281. Suggest replacing “spread” with “diffusion”.

15.   Page 10, line 282. APT-weighted imaging is sensitive to, but does not directly measure, the exchange rate of amide protons. This sentence would be fine if the word “rate” were removed. Note that this is the first time that the term “amide protons” has actually been used in the paper. The authors have assumed that the reader already knows that it is the amide protons in the proteins that APT-weighted imaging is sensitive to.

16.   Discussion is limited. Why do we need another method to guess lymph node status? What would this information do that we can’t do now? What makes it worth the larger price tag? Overall, this looks like a tool in search of an application and the authors don’t make a convincing case that this is the best application, or even a good application, for the tool.

Author Response

Response to Reviewer 3 Comments

Summary: The authors present a study of 61 patients with cervical cancer to examine the applicability of APT-weighted imaging and diffusion kurtosis imaging in predicting lymph node metastasis. They found that both methods correlated with lymph node status. However, they did not provide a hypothesis for their study, nor did they explain whether their finding would justify the increased imaging costs.

Major weaknesses of the paper include the fact that there is no hypothesis presented. The discussion focuses solely on the correlation between the imaging metrics and the lymph node status without addressing how useful either technique would be clinically. It is not clear how the lymph node status was determined. Assuming that the status was determined via pathology, it is not clear if the addition of these imaging methods would eliminate the need for pathological examination.

Specific comments:

Point 1: Page 2, line 54. ZOOMit is a Siemens-specific sequence. It would be helpful to explain what this sequence is in generic terms instead of using a vendor specific name, at least in the first mention.

Response 1: We thank the Reviewer’s suggestion. We have explained ZOOMit in generic terms in the introductory part (Page 2).

Point 2: Page 2, general observation. There is no hypothesis presented nor is there any mechanistic explanation for why APT-weighted imaging or diffusion kurtosis would be any better than straight diffusion-weighted imaging.

Response 2: We thank the Reviewer’s suggestion. Relevant explanation for why APT-weighted imaging or diffusion kurtosis would be any better than straight diffusion-weighted imaging has been given in the introduction. DKI has the potential in quantifying the microstructural heterogeneity of tissues and APT-weighted imaging provides more metabolic information than the changes of tissue microstructure. Both of them can reflect tissue information better than the diffusion coefficient of water molecules in tissue explored by DWI.

Point 3: Page 2, line 82. This is obviously a research sequence, so the patients should have provided written informed consent for this. Even if the current analysis was on previously obtained data, such that informed consent was not needed for this specific analysis, some mention of informed consent for the original image acquisition should be made.

Response 3: We agree with the Reviewer’s opinion. Our study was a retrospective study, and the ethics committee of our hospital approved this retrospective study, and the requirement for informed consent was waived. At present, we haven’t acquired the patient's written informed consent. We will pay attention to this problem in future studies.

Point 4: Page 3, line 101. This reference cites a Philips scanner, but the current paper uses a Siemens scanner. Thus, someone did a conversion from one vendor to the other, which should be indicated in the description. This wasn't a binary code that was provided for plug 'n' play.

Response 4: We thank the Reviewer for pointing out this issue. We changed a reference containing Siemens scanner. The CEST sequence we used was developed by Professor Zhang Yi's team from Zhejiang University, which has been well verified and applied in Siemens and Philips. Therefore, we use this sequence directly on the Siemens scanner.

Point 5: Page 3: lines 105-113. What sort of readout sequence was used with the CEST imaging? Was it a gradient recalled echo, a fast spin echo, or an echo-planar readout? If it was a fast spin echo, how long was the echo train? If it was an EPI readout, was it single shot? Was fat sat used?  Is the bandwidth cited the bandwidth for excitation or readout? Or was it the bandwidth of the CEST pulse? Does the acquisition time refer to the entire set of 63 offsets? If so, then this almost has to be an EPI readout. Given that so many frequency offsets were acquired, why did the authors only use the simple MTRasym analysis instead of any kind of fitting (e.g. Zaiss’s method, give reference).

Response 5: We thank the Reviewer’s suggestion. The CEST imaging used a fast spin echo. The echo train was 128. It was the bandwidth of CEST pulse. The acquisition time refers to the entire set of 63 offsets. We used the program prepared by Professor Zhang Yi's team from Zhejiang University, which only use the simple MTRasym analysis. In future study, we will further explore this problem.

Point 6: Page 3, line 118: Is this bandwidth for the excitation pulse or the readout? And is the acquisition time the total required for all 5 b-values? How many directions were used?

Response 6: We thank the Reviewer’s suggestion. This bandwidth is for the excitation pulse. The acquisition time is the total required for all 5 b-values. There are 3 directions were used.

Point 7: Page 4, figure 2. It would be helpful to outline the lesion on the T2 image and carry that outline across the other images. Why does the Z-spectrum run from 80 to -10 ppm? It would be much easier to assess the APT effect if the spectrum were only shown from about 8 to -8 ppm.

Response 7: We have taken the Reviewer’s suggestion and outlined the lesion on the T2 image and carried that outline across the other images (Page 4, Figure 2). We used the program prepared by Professor Zhang Yi's team from Zhejiang University, which directly generated Z-spectrum from 80 to -10 ppm. Professor Zhang Yi's team has successfully verified the application value of this sequence and program in previous studies[1-3].

  1. Liu, R.; Zhang, H.; Qian, Y.; Hsu, Y.C.; Fu, C.; Sun, Y.; Wu, D.; Zhang, Y. Frequency-stabilized chemical exchange saturation transfer imaging with real-time free-induction-decay readout. Magn Reson Med 2021, 85, 1322-1334, doi:10.1002/mrm.28513.
  2. Liu, R.; Zhang, H.; Niu, W.; Lai, C.; Ding, Q.; Chen, W.; Liang, S.; Zhou, J.; Wu, D.; Zhang, Y. Improved chemical exchange saturation transfer imaging with real-time frequency drift correction. Magn Reson Med 2019, 81, 2915-2923, doi:10.1002/mrm.27663.
  3. Zhang, Y.; Heo, H.Y.; Lee, D.H.; Zhao, X.; Jiang, S.; Zhang, K.; Li, H.; Zhou, J. Selecting the reference image for registration of CEST series. J Magn Reson Imaging 2016, 43, 756-761, doi:10.1002/jmri.25027.

Point 8: Page 5, figure 3. Same comments as for previous figure.

Response 8: We have taken the Reviewer’s suggestion and outlined the lesion on the T2 image and carried that outline across the other images (Page 5, Figure 3).

Point 9: Page 5, line 186. Delete “while”, as it makes this sentence incomplete.

Response 9: We have taken the Reviewer’s suggestion and deleted “while” in the sentence (Page 5).

Point 10: Page 7, line 224. Delete "while”.

Response 10: We have taken the Reviewer’s suggestion and deleted “while” in the sentence (Page 7).

Point 11: Page 7, lines 220-223. If all of these parameters (tumor size, histological grade, depth of invasion) correlate with lymph node status, what is the advantage of adding APT and DKI? This needs to be addressed somewhere and there should be a huge advantage to offset the increased cost. 

Response 11: We thank the Reviewer’s suggestion. Tumor size, histological grade, and depth of invasion are the only risk factors for lymph node metastasis. The larger the tumor, the higher the grade and the deeper the invasion will increase the risk of lymph node metastasis in cervical cancer. However, these clinicopathologic factors cannot be used as the criteria for predicting lymph node metastasis of cervical cancer in clinical practice. APT and DKI can not only provide structural information about the tumor, but also can provide the microenvironment and microstructure of the tumor. APT and DKI are supposed to quantitatively assess the tumor microenvironment, which may provide information to predict lymph node metastasis of cervical cancer with non-invasive detection methods.

Point 12: Figure 4 shows quite a bit of overlap between the groups for the various MRI parameters. What does this mean for an individual patient? This should be discussed.

Response 12: We agree with the reviewer's opinion that there is indeed a bit of overlap in the scatter plot in Figure 4, which we cannot deny. This is where our paper is not perfect, and we will pay more attention to this problem in the future research. Although our scatter plot shows a bit of overlap, which may not show a difference for individual patient, the difference is statistically significant when comparing the mean value between the two groups. This is valuable in reflecting the general trend. Moreover, relatively speaking, APT overlaps smaller areas than MK and MD, which further indicates that APT has a better effect in predicting lymph node metastasis of cervical cancer. This is consistent with the conclusion of our paper.   

Point 13: Page 10, line 270. What do the authors mean by “blocking degree”? MD should represent how freely water can diffuse through a tissue, with lower MD indicating impaired diffusion and probably denser tissue.

Response 13: We agree with the reviewer’s suggestion.The sentence has been modified with “MD represents how freely water can diffuse through a tissue, with lower MD indicating impaired diffusion and probably denser tissue.” (Page 10).

Point 14: Page 10, line 281. Suggest replacing “spread” with “diffusion”.

Response 14: We have taken the Reviewer’s suggestion and replaced “spread” with “diffusion” in the sentence (Page 10).

Point 15: Page 10, line 282. APT-weighted imaging is sensitive to, but does not directly measure, the exchange rate of amide protons. This sentence would be fine if the word “rate” were removed. Note that this is the first time that the term “amide protons” has actually been used in the paper. The authors have assumed that the reader already knows that it is the amide protons in the proteins that APT-weighted imaging is sensitive to.

Response 15: We have taken the Reviewer’s suggestion and removed the “rate” in the sentence (Page 10).

Point 16: Discussion is limited. Why do we need another method to guess lymph node status? What would this information do that we can’t do now? What makes it worth the larger price tag? Overall, this looks like a tool in search of an application and the authors don’t make a convincing case that this is the best application, or even a good application, for the tool.

Response 16: We thank the Reviewer for pointing out this issue. We already explained why do we need another method to guess lymph node status in the introduction. MRI has been established as the main imaging modality in pre-treatment assessment of lymph node metastasis in patients with cervical cancer. The application of the conventional MRI sequences, which are mainly based on morphologic features, is still challenging due to their low sensitivity (29%–69%) for assessing the presence of lymph node metastasis. DKI has the potential in quantifying the microstructural heterogeneity of tissues and CEST imaging provides more metabolic information than the changes of tissue microstructure. This suggests that APT and DKI may provide more information about the tumor than conventional MRI. The combination of APT and DKI is more valuable in predicting lymph node metastasis. Although APT has not been included in clinical application sequence, it has great potential to evaluate tumor microenvironment. Our study is only an exploratory study, showing that APT combined with DKI has the potential in improving the predictive efficacy of lymph node metastasis in cervical cancer. APT has a good development trend in research and clinical application.
